# Effects of Antiretroviral Treatment on Central and Peripheral Immune Response in Mice with EcoHIV Infection

**DOI:** 10.3390/cells13100882

**Published:** 2024-05-20

**Authors:** Qiaowei Xie, Mark D. Namba, Lauren A. Buck, Kyewon Park, Joshua G. Jackson, Jacqueline M. Barker

**Affiliations:** 1Department of Pharmacology and Physiology, Drexel University College of Medicine, Philadelphia, PA 19102, USA; qx33@drexel.edu (Q.X.); mdn54@drexel.edu (M.D.N.); lap352@drexel.edu (L.A.B.); jgj33@drexel.edu (J.G.J.); 2Graduate Program in Pharmacology and Physiology, Drexel University College of Medicine, Philadelphia, PA 19102, USA; 3Center for AIDS Research, University of Pennsylvania, Philadelphia, PA 19104, USA; kyewpark@pennmedicine.upenn.edu

**Keywords:** EcoHIV, HIV, mouse model, immune response, nucleus accumbens, microglia, cytokine, ART

## Abstract

HIV infection is an ongoing global health issue, despite increased access to antiretroviral therapy (ART). People living with HIV (PLWH) who are virally suppressed through ART still experience negative health outcomes, including neurocognitive impairment. It is increasingly evident that ART may act independently or in combination with HIV infection to alter the immune state, though this is difficult to disentangle in the clinical population. Thus, these experiments used multiplexed chemokine/cytokine arrays to assess peripheral (plasma) and brain (nucleus accumbens; NAc) expression of immune targets in the presence and absence of ART treatment in the EcoHIV mouse model. The findings identify the effects of EcoHIV infection and of treatment with bictegravir (B), emtricitabine (F), and tenofovir alafenamide (TAF) on the expression of numerous immune targets. In the NAc, this included EcoHIV-induced increases in IL-1α and IL-13 expression and B/F/TAF-induced reductions in KC/CXCL1. In the periphery, EcoHIV suppressed IL-6 and LIF expression, while B/F/TAF reduced IL-12p40 expression. In the absence of ART, IBA-1 expression was negatively correlated with CX3CL1 expression in the NAc of EcoHIV-infected mice. These findings identify distinct effects of ART and EcoHIV infection on peripheral and central immune factors and emphasize the need to consider ART effects on neural and immune outcomes.

## 1. Introduction

Worldwide, there are approximately 39 million people living with HIV (PLWH), 75% of whom are accessing antiretroviral therapy (ART) [1]. HIV infection is associated with immune responses that include increased circulating cytokine levels in the periphery and within the central nervous system (CNS), which may contribute to the severity of the disease [2]. Although ART can effectively maintain the HIV-1 viral load at undetectable levels and prolong the lifespan of PLWH, ART does not completely eliminate the infection or the impact of HIV-induced immune response peripherally or within the central nervous system (CNS) on comorbidities including HIV-associated neurocognitive disorder (HAND) [3,4,5].

HIV-1 enters the CNS via infected peripheral monocytes and T cells, which can cross the blood–brain barrier [6,7,8]. Once within the CNS, HIV can infect glial cells including microglia and astrocytes—but not neurons—establishing a viral reservoir that persists, despite ART treatment [9,10,11]. The presence of HIV and HIV proteins in the brain can induce the release of inflammatory factors, contributing to neuroinflammation and neuronal dysfunction [12,13]. These deficits may be further exacerbated within reward-related substrates, as findings from both clinical populations and preclinical models identify deficits within the nucleus accumbens (NAc) and its connecting structures that may ultimately contribute to the aberrant reward processing, learning, and memory seen in PLWH [14,15,16,17]. For example, HIV dampens dopamine (DA) system function and dysregulates glutamatergic synaptic plasticity in the NAc, likely through perturbations in immune signaling [18,19]. Considering the high rates of comorbid substance use disorder (SUD) and HIV infection, investigating neuroimmune changes in the NAc may provide insight into the development of novel approaches to mitigate addiction-related behaviors and neurocognitive outcomes in PLWH with SUD.

Beyond the persistence of the inflammatory response in PLWH, even in the presence of ART, it is becoming increasingly evident that ART interacts with HIV and/or acts independently to produce a distinct immune profile in the periphery and CNS [20,21,22,23,24]. In clinical findings, the direct effects of ART are difficult to disentangle from interactions with aging, viral status, and additional comorbidities. However, preclinical models point to the ART-induced dysregulation of peripheral and central nervous system immune function [25]. Cognitive outcomes include spatial learning and memory impairments following treatment with nevirapine [26] or emtricitabine + tenofovir disoproxil fumarate [25] in the absence of infection. Thus, preclinical research will be essential to fully characterize the independent and interactive effects of ART on neural and immune outcomes.

Together, these findings point to the need to understand the independent and interactive effects of HIV infection and ART on peripheral and CNS immune outcomes, necessitating preclinical models of HIV [18,27,28,29]. To accomplish this, the current study utilized the EcoHIV infection mouse model, in which a chimeric virus is generated by replacing the coding region of the HIV-1 glycoprotein gp120 with that of gp80 [30]. Here, infection was initiated in the periphery. To model a commonly prescribed ART regimen, the current study used a daily combination treatment with bictegravir (B)—a recommended integrase strand transfer inhibitor (INSTI)—emtricitabine (F), and tenofovir alafenamide (TAF)—nucleoside reverse transcriptase inhibitors (NRTIs)—in accordance with recommended initial ART regimens that show high rates of viral suppression in PLWH [31,32].

Experiments in this study assessed immune factors in plasma and in the NAc, a key substrate of brain reward circuitry known to be dysregulated in models of HIV [16,29], including following CNS infection with EcoHIV [33]. Neuroimmune dysregulation in this region can impair excitatory glutamatergic neuronal plasticity, thus driving increased susceptibility to addictive drugs [34,35]. Given the evidence that microglia are the principal HIV-1 reservoir within the CNS, as well as the known role of microglia in sensing neural activity and modulating neuronal plasticity [11], we investigated EcoHIV and ART effects on microglia. Fractalkine (CX3CL1) is a soluble chemokine released by neurons, it is an exclusive ligand for CX3CR1 receptors that are expressed on microglia [36]. CX3CL1/CX3CR1 signaling has been proposed to mediate neuron–microglia communication that underlies microglial activation and neuroimmune response [37,38] and thus we further characterized the NAc expression of Iba-1 and CX3CL1. The present results identify independent and interactive effects of both EcoHIV infection and B/F/TAF treatment on cytokine expression in both plasma and the NAc.

## 2. Materials and Methods

### 2.1. Subjects

Adult male (n = 24) and female (n = 24) C57BL/6J mice (9 weeks old upon arrival) were obtained from Jackson Laboratories (Bar Harbor, ME, USA). Following arrival, mice were group housed in same-sex cages for 7 days to acclimatize, with ad libitum access to a standard chow diet and water. Mice were housed at the Drexel University College of Medicine under standard 12 h light:12 h dark conditions in microisolation conditions. All experiments were approved by the Institutional Animal Use and Care Committee at Drexel University.

### 2.2. EcoHIV Virus Generation

Plasmid DNA encoding the EcoHIV-NDK coding sequence (gift from Dr. David Volsky) was purified from bacterial stocks (Stbl2 cells #10268019, Thermo Fisher, Waltman, MA, USA), using an endotoxin-free plasmid purification kit (ZymoPure #D4200, Zymo Research, Irvine, CA, USA). Purified DNA was transfected into nearly confluent (80–90%) 10 cm^2^ plates of low passage LentiX 293T (#632180, Takara Bio, San Jose, CA, USA), using a calcium phosphate transfection protocol. The cell culture supernatant was collected at 48 h post-transfection and cellular debris was pelleted using centrifugation at low speed (1500× *g*) on a benchtop centrifuge (4 °C), followed by passage through a cell strainer (40 µm). The supernatant, containing viral particles, was mixed 4:1 with a homemade lentiviral concentrator solution (4×; MD Anderson) composed of 40% (*w*/*v*) PEG-8000 and 1.2 M NaCl in PBS (pH 7.4). The supernatant–PEG mixture was placed on an orbital shaker (60 rpm) and incubated overnight at 4 °C. This mixture was centrifuged at 1500× *g* for 30 min at 4 °C. After centrifugation, the medium was removed and the pellet was rinsed 1× with PBS. The pellet was centrifuged again (1500× *g* for 5 min), the solution was removed, and the viral pellet was resuspended in cold, sterile PBS. The viral titer (p24 core antigen content) was determined initially using a LentiX GoStix Plus titration kit (#631280, Takara Bio, San Jose, CA, USA) and subsequently using an HIV p24 AlphaLISA detection kit (#AL291C, PerkinElmer, Waltman, MA, USA). Viral stocks were aliquoted and stored at −80 °C until used.

### 2.3. EcoHIV Inoculation and Antiretroviral Drug Self-Administration

Following 1 week of acclimation, mice were matched based on sex into EcoHIV or sham control groups. Mice were inoculated with EcoHIV at a dose of 300 ng p24 EcoHIV-NDK, i.p. For inoculation, mice were lightly anesthetized with 5% isoflurane and injected interperitoneally with either EcoHIV or vehicle (sterile 1x PBS) for sham controls and were, subsequently, individually housed.

Antiretroviral drug powder (bictegravir, emtricitabine, and tenofovir alafenamide, B/F/TAF, Cayman Chemicals, Ann Arbor, MI, USA), was pre-mixed with Rodent Liquid Diet (Cat#F1268, AIN-76, Bio-Serv, Flemington, NJ, USA) as a liquid mixture. Compounds were stored and prepared in a manner consistent with manufacturer guidance and were used within the expected efficacy span. Mice were assigned based on sex and infection status into B/F/TAF-treated and vehicle-treated groups. Beginning one week after inoculation, EcoHIV-infected and sham control mice were single-housed for the control of daily B/F/TAF dosing via oral self-administration. This time point was selected as it occurs after the initial escalation of EcoHIV viral RNA [39] and thus was not expected to impact the establishment of infection. Mice consumed 5 mL B/F/TAF mixture, dosed as 0.375 mg B/1.23 mg F/0.15375 mg TAF per mouse diet per day (calculated as the human equivalent dose [40,41,42]), or received 5 mL liquid diet, as vehicle controls, daily until the end of the test. To ensure controlled dosage consumption, mice were subjected to food restriction, and additional food pellet increments were administered as needed to maintain body weights at pre-restriction levels.

### 2.4. Brain and Blood Sample Collection and Preparation

#### 2.4.1. Blood Samples

Blood samples were collected from all mice 1, 3, and 5 weeks following EcoHIV inoculation. For blood collections, mice were briefly anesthetized with 5% vaporized isoflurane and, using the submandibular collection method, blood was collected from each mouse. Blood samples were centrifuged at 8700× *g* for 20 min. Plasma was collected from supernatant and stored at −80 °C. Following blood collection at the end of week 5, mice were overdosed on isoflurane and euthanized via rapid decapitation, brains and spleens were removed, flash-frozen on dry ice, and stored at −80 °C.

#### 2.4.2. Nucleus Accumbens (NAc) Lysate

Frozen brains were sectioned and bilateral NAc was isolated using a biopsy punch and collected into 1.5 mL tubes. NAc brain tissues were homogenized in RIPA buffer (Santa Cruz Biotechnology, Dallas, TX, USA) and incubated at 4 °C for 30 min. In total, 1% Nonidet P-40 was used as the detergent in the RIPA buffer, to inactivate the virus for biosafety reasons; this was confirmed by the vendor to be compatible with the assay. Homogenized NAc samples were centrifuged at 12,000× *g* for 10 min. The supernatants were collected for whole tissue lysate. Protein concentration was quantified using Pierce™ BCA Protein Assay Kit (Thermo Scientific, Waltham, MA, USA).

### 2.5. EcoHIV Infection Validation

Spleens were harvested at the time of euthanasia and were flash-frozen for storage at −80 °C. DNA was isolated from spleens using the Qiagen QIAamp DNA Mini Kit (#51304, Qiagen, Germantown, MD, USA). Confirmation of infection was performed by the University of Pennsylvania Center for AIDS Research (CFAR). The CFAR received DNA samples for a cell-associated DNA assay and performed qRT-PCR using the HIV-LTR primers and probe provided below. The OD was used to estimate the input cell numbers to normalize the data.

Kumar LTR F, GCCTCAATAAAGCTTGCCTTGA

Kumar LTR R, GGGCGCCACTGCTAGAGA

Kumar LTR Probe (FAM/BHQ), 5′ CCAGAGTCACACAACAGACGGGCACA 3′

### 2.6. Quantification of NAc and Peripheral Immune Proteins using the Mouse Cytokine 32-PlexDiscovery Assay

Cytokine and chemokine expression in NAc lysates and plasma were analyzed using a multiplex immunoassay. Mouse NAc lysates were diluted 1:1 in PBS (PH ~ 7.5) prior to shipping. Plasma from week 5 was diluted 1:1 in PBS. A total of 1% Triton was added to the plasma to inactivate the virus for shipment; this was confirmed by the vendor to be compatible with the assay. Cytokines, chemokines, and growth factors in NAc lysate and plasma were measured using the Mouse Cytokine 32-PlexDiscovery Assay (Eve Technologies, MD32, Calgary, AB, Canada). The multiplex analysis was performed using the Luminex 200 system (Luminex, Austin, TX, USA) from Eve Technologies Corporation. Results that were out of range (i.e., outside the standard curve) were excluded from the dataset prior to analysis. The assay sensitivities of the mouse markers range from 0.3 to 30.6 pg/mL. The 32 targets include: Eotaxin, G-CSF, GM-CSF, IFNγ, IL-1α, IL-1β, IL-2, IL-3, IL-4, IL-5, IL-6, IL-7, IL-9, IL-10, IL-12p40, IL-12p70, IL-13, IL-15, IL-17A, IP-10, KC, LIF, LIX, MCP-1, M-CSF, MIG, MIP-1α, MIP-1β, MIP-2, RANTES, TNFα, and VEGF.

### 2.7. Western Blot

Aliquots of NAc lysate from the same samples used for the cytokine assay were analyzed for Iba-1 and CX3CL1 expression via Western blot. The NAc tissue was lysed into whole tissue lysate in RIPA buffer containing protease inhibitor, clarified using centrifugation, and the protein concentrations were determined using a BCA Protein Assay. Equal amounts of protein (12 μg) were loaded and run on 4–12% gradient gel (NuPAGE^TM^, 4–12%, Bis-Tris, 1.0 mm). Proteins were transferred onto nitrocellulose membrane using an iBlot^TM^ 2 Gel Transfer Device. The membranes were then blocked using 5% BSA (Iba-1) or 5% milk (CX3CL1) in Tris-buffered saline + 0.1% Tween-20 (TBS-T) for 2 h. Rabbit anti-Iba-1 antibody (1:500; 019-19741, FUJIFILM Wako Chemicals, Richmond, VA, USA), rabbit anti-CX3CL1 (1:500; bs-0811R, Bioss, Woburn, MA, USA), or rabbit anti-GAPDH antibody (1:3000; D16H11; Cell Signaling Technology, Danvers, MA, USA) was added to the blocking buffer, and membranes were incubated overnight at 4 °C. The membranes were then washed in TBS-T and incubated with horseradish peroxidase-labeled (HRP) goat anti-rabbit secondary antibody (Abcam ab97080; 1:3000 for Iba-1 and CX3CL1, 1:10,000 for GAPDH) for 1 h. A chemiluminescent substrate (SuperSignal™ West Pico Plus, Thermo Fisher, Waltham, MA, USA) was used to develop the membranes, which were measured using a Licor Odyssey^®^ FC imager. Protein bands were analyzed using ImageJ (FIJI v2.15.1). A between-gel control sample was loaded onto every gel and the expression of Iba1 and CX3CL1 was normalized to this sample, prior to normalizing to GAPDH to control for between-gel differences. Thus, data were calculated as ratios of normalized Iba-1 or CX3CL1/normalized GAPDH and then as a fold change relative to the mean of the control group (vehicle mice).

### 2.8. Statistical Analyses

All data were analyzed in GraphPad Prism. Body weights were analyzed using a three-way mixed model ANOVA using Greenhouse–Geisser correction. Cytokine/chemokine data were analyzed using a two-way ANOVA or an unpaired *t*-test. Analyses were Log2 transformed or square root transformed when normal distribution was not met. Significant interactions were followed using Tukey’s post hoc analysis. Correlational analyses were performed using simple linear regression. Significance levels for each test were at *p* < 0.05. Principal component analysis (PCA) was applied to identify common characteristics among the dependent variables using Prism. Raw expression data of cytokine/chemokine were scaled and centered. The correlation matrix and eigenvalue were calculated, eigenvalues greater than 1 were considered significant in contributing to the components, and data were plotted on the first two components. A z-score two population comparison was applied to compare the proportion of mice assigned to each quadrant following PCA.

## 3. Results

### 3.1. EcoHIV-NDK Infection of Wild Type Mice

Adult male and female C57BL/6J mice (9 weeks) were matched by sex to undergo either sham or EcoHIV inoculation and vehicle or B/F/TAF exposure (Figure 1A). To confirm EcoHIV-NDK infection status, terminal viral DNA burden was measured in spleens using qPCR. A subset of tissues (n = 7) was run from sham control mice to confirm that no viral burden was observed. EcoHIV-NDK-infected mice showed viral burdens in the range from 0.3 to 5 × 10^3^ viral DNA copies per 10^6^ spleen cells, after 5 weeks of EcoHIV infection. One EcoHIV vehicle-treated mouse was excluded from analyses as no viral DNA was detected. There was no significant difference in mean EcoHIV DNA viral burden between B/F/TAF-treated mice (n = 12) and vehicle-treated EcoHIV mice (n = 11) [t (21) = 0.103, *p* = 0.9191; Figure 1B]. Body weight was monitored 3 times per week following inoculation (Figure 1C). No effects of EcoHIV infection [F (1, 20) = 1.325, *p* = 0.2633] or B/F/TAF treatment [F (1, 20) = 0.8443, *p* = 0.3691] were observed on change in weight. However, there was an effect of time [three-way ANOVA; F (2.006, 40.12) = 37.13, *p* < 0.0001; Greenhouse–Geisser corrected, Figure 1C]. A post hoc Dunnett’s test revealed that weights were reduced on day 3 compared to day 0 (*p* < 0.05). Weights were greater than day 0 on day 24, 27, 30, and 33 (*p*’s < 0.05).

### 3.2. EcoHIV Infection Alters Neuroimmune Responses in the NAc

To assess the expression levels of immune markers in the brain of EcoHIV-infected mice, this study utilized the mouse cytokine 32-PlexDiscovery Assay. Results for analytes that were outside the linear range of detection were excluded from the dataset prior to analysis (Table 1). Two-way ANOVA revealed changes in immune markers. EcoHIV independently increased the expression of several targets, including IL-1α [main effect of EcoHIV: F (1, 19) = 0.1221, *p* = 0.0479; no main effect of B/F/TAF: F (1, 19) = 0.1221, *p* = 0.7306; no interaction: F (1, 19) = 3.315, *p* = 0.0845; Figure 2A] and IL-13 [main effect of EcoHIV: F (1, 19) = 4.520, *p* = 0.0468; no main effect of B/F/TAF: F (1, 19) = 0.04333, *p* = 0.8373; no interaction: F (1, 19) = 1.096, *p* = 0.3083; Figure 2B]. Interestingly, B/F/TAF treatment independently decreased keratinocyte chemoattractant (KC; i.e., CXCL1) expression [main effect of B/F/TAF: F (1, 19) = 6.170, *p* = 0.0225; no main effect of EcoHIV: F (1, 19) = 2.222, *p* = 0.1525; no interaction: F (1, 19) = 1.919, *p* = 0.1820; Figure 2C].

Two-way ANOVAs revealed significant interactions between EcoHIV infection and B/F/TAF treatment for IFNγ, IL-7, IL-10, and RANTES. For IFNγ (Figure 2D), a significant interaction [F (1, 19) = 5.144, *p* = 0.0352], but no main effects of EcoHIV [F (1, 19) = 0.01158, *p* = 0.9154] or B/F/TAF [F (1, 19) = 0.3436, *p* = 0.5647], were observed. However, post hoc analyses revealed no significant comparisons. For IL-7 (Figure 2E), a significant interaction [F (1, 19) = 4.917, *p* = 0.0390] and significant main effects of EcoHIV [F (1, 19) = 5.683, *p* = 0.0277] and B/F/TAF [F(1, 19) = 4.893, *p* = 0.0394] were observed. A post hoc analysis revealed IL-7 was significantly decreased in EcoHIV mice treated with B/F/TAF, compared to all other groups (* *p*’s < 0.05). For IL-10 (Figure 2F), a significant interaction [F (1, 19) = 7.397, *p* = 0.0136] and main effect of EcoHIV [F (1, 19) = 4.414, *p* = 0.0492], but no significant main effect of B/F/TAF [F (1, 19) = 0.7504, *p* = 0.3972], were observed. A post hoc analysis indicated that, in vehicle-treated mice, EcoHIV significantly enhanced IL-10 vs. sham-inoculated mice (post hoc: *p* = 0.0169). There was no difference between groups treated with B/F/TAF (*p*’s > 0.1). Similar to IFNγ, analysis of RANTES expression revealed a significant interaction [F (1, 19) = 6.197, *p* = 0.0222] but no main effects of EcoHIV [F (1, 19) = 0.8639, *p* = 0.3643] or B/F/TAF [F (1, 19) = 0.4738, *p* = 0.4995; Figure 2G] or significant post hoc comparisons. As the NAc expression of all analytes was not normally distributed, all data were Log2 transformed prior to two-way ANOVA. The value for each analyte (uncorrected), represented as the fold-changes versus the parent control group (sham infection, vehicle treated), are visualized in the heatmap in Appendix A (EcoHIV-veh: n = 5; sham-B/F/TAF: n = 6; EcoHIV-B/F/TAF: n = 6; * *p* < 0.05). Values were compared using an unpaired *t*-test and the exact *p* values are presented in Appendix A.

To further characterize the neuroimmune markers that were associated with EcoHIV and B/F/TAF-treatment groups, we applied principal component analysis (PCA), a dimension reduction approach, using cytokines/chemokines that were significantly modulated in individual analyses (Figure 3A). The first two principal components captured the majority of the variance (71%). Distribution along the first principal component (PC1) was driven by IL-1α, IFNγ, IL-10, and RANTES (“Neuroimmune Cluster”), explaining 52.9% of the cumulative variance. The relationship between each marker was highly correlated. The second principal component (PC2) accounted for 19% of the variance and was positively correlated with IL-13 and negatively correlated with IL-7. The vector plot is shown in Appendix A. Minimally overlapping biomarker distributions were observed between EcoHIV-infected and sham mice, with distribution along the PC2 axis corresponding with EcoHIV infection status (EcoHIV was associated with High IL-13, sham was associated with High IL-7). Distribution along PC1 corresponded with B/F/TAF treatment. Assignment to each quadrant was not equally distributed within groups of mice. In particular, 100% of vehicle-treated EcoHIV-infected mice were in the High IL-13 and High Neuroimmune quadrant. Treatment with B/F/TAF reduced the proportion of mice in this quadrant, with 33.33% of B/F/TAF-treated EcoHIV mice in this quadrant (z-score two population comparison; z = 2.2887, *p* = 0.0220; Figure 3B). The majority of the remaining B/F/TAF-treated EcoHIV mice (50% of total) were in the High IL-13 and Low Neuroimmune cluster. No sham-infected mice were in the High IL-13 and High Neuroimmune cluster and only one was within the High IL-13 category. This distinct cytokine distribution indicated EcoHIV infection shifted the cytokine profile toward high neuroimmune responses and B/F/TAF treatment attenuated this effect in EcoHIV-infected mice.

### 3.3. EcoHIV Infection Alters Peripheral Inflammatory Responses in Plasma

To determine the peripheral inflammatory response associated with EcoHIV infection and ART, plasma was analyzed in the 32-multiplex analysis. Two cohorts of samples, including both sham- and EcoHIV-infected mice, were included in this analysis. To minimize the impact of assay batch differences, the values for each analyte were represented as the fold-change from the mean of the sham-veh group within each cohort. Log_2_ transformation or square root transformation are applied to correct analytes that were not normally distributed (Table 2).

Findings indicated that, independent of B/F/TAF treatment, EcoHIV infection reduced the expression of IL-6 and LIF in plasma, compared to sham mice [IL-6 (Figure 4A): main effect of EcoHIV: F (1, 39) = 4.804, *p* = 0.0344; no main effect of B/F/TAF: F (1, 39) = 0.8515, *p* = 0.3618; no interaction: F (1, 39) = 0.8436, *p* = 0.3640; LIF (Figure 4B): main effect of EcoHIV: F (1, 34) = 4.768, *p* = 0.0360; no main effect of B/F/TAF: F (1, 34) = 0.001303, *p* = 0.9714; no interaction: F (1, 34) = 0.006513, *p* = 0.9362]. For IL-12p40 (Figure 4C), a main effect of B/F/TAF treatment was observed, such that IL-12p40 expression was lower in B/F/TAF-treated mice [main effect of B/F/TAF: F (1, 37) = 4.728, *p* = 0.036; no main effect of EcoHIV: F (1, 37) = 0.1134, *p* = 0.7382; no interaction: F (1, 37) = 0.5555, *p* = 0.4608].

Expression of IL-5 (Figure 4D) was driven by interactions between EcoHIV infection and B/F/TAF treatment [interaction: F (1, 43) = 4.080, *p* = 0.0497; no main effect of EcoHIV: F (1, 43) = 3.500, *p* = 0.0682; no main effect of B/F/TAF: F (1, 43) = 1.202, *p* = 0.2791]. Post hoc analysis revealed a significant reduction in IL-5, due to B/F/TAF treatment in EcoHIV-infected mice relative to uninfected mice treated with B/F/TAF (post hoc, *p* = 0.0385), but no differences were observed compared to the sham-Veh and EcoHIV-Veh groups.

### 3.4. EcoHIV Altered the Relationship between NAc Iba-1 Expression and Immune Factor Expression

To assess the effect of EcoHIV and B/F/TAF on putative microglia activation and CX3CL1 level, aliquots of the NAc lysate that were used for cytokine assays were analyzed using Western blot for Iba-1 and CX3CL1 expression. Normalized Iba-1 expression was represented here as the fold change relative to the mean of the sham-veh group of each cohort. No main or interaction effects of EcoHIV and B/F/TAF on NAc Iba1 expression were observed (Figure 5A) [Two-way ANOVA, n = 12/group; no main effect of EcoHIV: F (1, 42) = 2.729, *p* = 0.1060; no main effect of B/F/TAF: F (1, 42) = 0.2158, *p* = 0.6446; no interaction: F (1, 42) = 1.156, *p* = 0.2883]. Based on our a priori hypothesis that EcoHIV infection would increase microglia reactivity, Iba-1 levels between the sham-Veh and EcoHIV-Veh groups were compared. An unpaired *t*-test demonstrated a significant increase in Iba-1 expression in EcoHIV-infected, vehicle-treated mice compared to sham, vehicle-treated mice [t (21) = 2.168, *p* = 0.0418, 95% C.I. = 0.01390 to 0.6687; Appendix A], indicating that 5 weeks of EcoHIV infection significantly upregulated Iba-1 expression in the NAc. No effects of EcoHIV or B/F/TAF treatment were observed on CX3CL1 expression in the NAc [EcoHIV: F (1, 42) = 0.0007631, *p* = 0.9781; B/F/TAF: F (1, 42) = 0.05451, *p* = 0.8165; interaction: F (1, 42) = 0.09385, *p* = 0.7609; Appendix A]. No differences were observed in CX3CL1 expression between sham-Veh and EcoHIV-Veh mice [t (21) = 0.1846, *p* = 0.8553, 95% C.I. = −0.3007 to 0.2517; Appendix A]. Representative blots are shown in Figure 5C (full blots in Appendix A).

Previous studies have shown that microglia activation is negatively associated with fractalkine system expression. Thus, we performed simple linear regression analysis between Iba-1 and CX3CL1 in sham and EcoHIV vehicle-treated groups. Linear regression showed a negative correlation with Iba-1 and CX3CL1 levels in EcoHIV-Veh mice (R^2^ = 0.4825, *p* = 0.0177; Figure 5D), whereas no correlation was observed between Iba-1 and CX3CL1 in the sham-Veh group (R^2^ = 0.03419, *p* = 0.5651; Figure 5E). This result may indicate that EcoHIV-induced increases in Iba1 expression are at least partially suppressed by CX3CL1 signaling within the NAc.

To determine whether Iba-1 expression was associated with the NAc expression of immune factors, simple linear regression analysis was performed (Appendix A). In particular, we observed differential patterns of relationships in EcoHIV-infected mice that were not treated with B/F/TAF, especially the inversion of the relationship between CX3CL1 expression and Iba-1, LIF, and Iba-1. These correlations are summarized in Table 3.

## 4. Discussion

The current findings demonstrate independent and interactive effects of EcoHIV infection and ART exposure on peripheral and brain immune response in mice. Given previous findings indicating the sensitivity of brain reward circuitry to perturbation in models of HIV infection, these experiments focused on dysregulation within the NAc, where the increased expression of IL-1α and IL-13 were induced by EcoHIV infection. Within the NAc, treatment with the ART combination B/F/TAF independently reduced the expression of CXCL1/keratinocyte-derived chemokine (KC) and interacted with EcoHIV infection to alter the expression of IL-7, IL-10, IFNγ, and RANTES. Changes in chemokine expression in the NAc were not accompanied by gross alterations in Iba-1 expression, a putative marker of microglia, though the relationship between Iba-1 and chemokine expression was determined by EcoHIV and B/F/TAF treatment status. Notably, the immune targets for which alterations were observed in plasma were distinct from those observed within the NAc. In plasma, EcoHIV infection reduced the expression of IL-6 and leukemia inhibitory factor (LIF), a member of the IL-6 family. Treatment with B/F/TAF reduced the expression of IL-12p40 and interacted with the EcoHIV infection to selectively reduce IL-5 expression in EcoHIV-infected mice.

Neuroimmune response is an important CNS outcome following HIV infection, which may have deleterious effects on neural function through protracted inflammation. Neuroimmune dysregulation within the CNS can result from the infiltration of HIV-1-infected macrophages across the blood–brain barrier and the subsequent infection of resident CNS immune cells that maintain persistent CNS viral reservoirs [8,43,44]. HIV-1-infected macrophages and microglia release viral proteins and inflammatory cytokines/chemokines, which, in turn, dysregulates immune signaling and disrupts homeostatic neuronal function [7,45,46]. HIV infection is associated with interleukin-1 (IL-1) family regulation in the CNS [2,47,48,49]. A large majority of studies on HIV infection and the IL-1 family have focused on the IL-1β as a major mediator of HIV-associated neuroinflammation [50]. For example, in rats treated with HIV-1 gp120, IL-1β upregulation within the cortex is associated with neuronal dysfunction and cognitive deficits [51]. IL-1α is a key pro-inflammatory cytokine in the IL-1 family and it is upregulated in models of co-occurring HIV (Tat protein model) and drug exposure [52]. Elevated IL-1α is further observed in the cerebrospinal fluid of PLWH and is positively associated with a risk of cognitive impairment [53]. We found that IL-1α was upregulated in EcoHIV-infected mice, despite ART treatment, consistent with persistent elevations of CNS IL-1α, even in virally suppressed PLWH.

EcoHIV infection also increased IL-13 in the NAc. The function of IL-13 is highly context dependent and may have a neuroprotective/homeostatic function depending on the models [54,55,56]. Peripheral IL-13 administration also appears to be neuroprotective, such as by inducing anti-inflammatory, microglia-mediated immune responses [55]. In the CNS, IL-13 upregulation also promotes synaptic plasticity through actions on glutamate receptors. While this may protect neurons from excitotoxic death following brain injury [56], this effect could produce cognitive impairments akin to those observed in HAND [57]. Increased IL-13 release from reactive microglia has also been shown to be neurotoxic following lipopolysaccharide injection, which was further associated with astrocyte damage and blood–brain barrier (BBB) disruption [58]. Thus, it is essential to consider the function of IL-13 in the context of specific disorders and disease states, including the state of disease progression.

Elevated IL-1α and IL-13 expression in the NAc—key neural substrates of reward learning and processing—of EcoHIV-infected mice may be associated with excitatory neuronal adaptation. IL-1, released from reactive microglia in a drug addiction model, is involved in enhancing glutamate transmission through downregulating glutamate transporter (GLT-1) [59] and potentiating the activation of ionotropic glutamate receptors (e.g., AMPA and NMDA) in the NAc [60,61]. IL-1 and IL-13 effects on synaptic plasticity are thought to be associated with long-term potentiation of excitatory glutamatergic neurons [56,60,62,63], which overlap with the action of addictive substances in the CNS. Thus, the exposure of neurons to high IL-1α and IL-13 may induce differential neuronal susceptibility to excitatory glutamate stimulation. This may suggest one mechanism by which PLWH are at greater risk for the development of SUD when exposed to addictive drugs. This finding is informative to research on HIV-associated comorbidities, including SUD, as the number of PLWH with SUD continues to increase [18,64].

Interestingly, a selective reduction in NAc IL-7 was observed in B/F/TAF-treated mice infected with EcoHIV. IL-7 effects are mediated through binding to the receptor IL-7R. IL-7/IL-7R signaling is a critical regulator of CD4+ T cell homeostasis [65]. The HIV infection disruption of intracellular signaling downstream of IL-7R may contribute to the subsequent loss of CD4s, which could thus worsen immune activation [66,67]. Elevated levels of serum or plasma IL-7 are observed in PLWH and animal models [68,69]; IL-7 administration promotes HIV viral persistence during ART treatment, by promoting rapid viral production [65]. One potential mechanism that may contribute to the ART-mediated rescue of CD4+ and CD8+ T cells counts is through enhanced IL-7 clearance, via the increased receptor availability of IL-7R on T cells [70]. Altogether, these findings point toward a hypothesis that increases in IL-7, the dysregulation of IL-7/IL-7R signaling may contribute to the pathophysiology of HIV, and that reduced IL-7 may help mediate the therapeutic efficacy of ART. In the present study, IL-7 was only decreased in EcoHIV-infected mice with B/F/TAF treatment. In addition to IL-7, interactions between EcoHIV infection and B/F/TAF were observed in IFNγ, IL-10, and RANTES, which are associated with neuropathogenesis and the persistence of HIV-associated neurocognitive deficits [53,71,72]. Specifically, NAc IL-10 expression was significantly greater in vehicle-treated EcoHIV-infected mice to Sham-Veh controls. Upregulated IL-10 has also been reported to play a role in viral load persistence and T cell impairment in the context of HIV [73,74].

PCA analysis indicated that EcoHIV infection promoted, and ART treatment attenuated, neuroimmune response. Vehicle-treated EcoHIV mice were distributed in the quadrant associated with High IL-13 and High Neuroimmune response, while treatment with B/F/TAF shifted the profile of EcoHIV-infected mice to a lower Neuroimmune response, without changing IL-13 levels. Notably, no sham-infected animals were in the High-IL-13/High Neuroimmune response quadrant. Further, B/F/TAF treatment in sham mice did not have the same effects on immune profile, consistent with the independent and interactive effects of ART treatment on immune status in the CNS.

The plasma immune profile is widely used as a biomarker for monitoring HIV disease progression. Peripheral immune activation in response to HIV is essential for protective immunity against viral invasion. In general, plasma cytokine dysregulation during acute and chronic HIV infection is associated with CD4+ and CD8+T cell activation status, viral load set-point, and the efficacy of ART [75,76]. Notably, in the current study, the observed changes in immune profiles were not congruent between plasma and the NAc (Appendix A). In EcoHIV-infected mice, downregulated plasma IL-6 and LIF expression was observed, regardless of B/F/TAF treatment. IL-6 is a proinflammatory cytokine that is elevated by HIV-infected monocytes and macrophages [77]. Increased circulating IL-6 levels are linked to HIV replication, decreased CD4+ cell counts, and elevated inflammatory response in PLWH [78]. A blockade of IL-6 activity has been reported to attenuate these effects [79], indicating that IL-6 can serve as both a marker for HIV inflammation outcome and plays a role in reversing these outcomes. Leukemia inhibitor factor (LIF) may be playing a similar role in the suppression of HIV-1 replication [80,81,82]; the loss of LIF may contribute to HIV transmission and increased viral load [80]. These findings are consistent with the interpretation that B/F/TAF treatment is not only interacting with EcoHIV infection to alter outcomes, as observed for IL-7 in the NAc or IL-5 in the periphery, but—similar to CXCL1/KC findings within the NAc—is independently altering immune targets, even in the absence of infection.

Within the CNS, HIV mRNA is thought to be predominantly expressed in microglia. High co-localization between Iba1 and HIV-1 mRNA has been observed in the HIV-1 Tg rats model [33]. In the EcoHIV model following an intracranial or retro-orbital route of infection—a different route than the peripheral inoculation method used in the current study—EcoHIV-EGFP signal was highly co-localized with Iba-1 in the CNS [33]. In addition, intracranial EcoHIV infection induced reactive morphology in microglia, increased expression of inflammatory factors, and neuronal and synaptic impairments (but not apoptosis) in the basal ganglia [83]. These results align with our findings that, following 5 weeks of EcoHIV infection, independent of B/F/TAF treatment, immune activation in the NAc was increased (including IL-1α, IL-10, and IL-13), potentially driven by activated microglia. This suggests that EcoHIV may facilitate microglia activation and associated immune responses in the NAc as early as 5 weeks post-infection. Although the intracerebral EcoHIV injection model showed a drastic change in Iba-1 expression and immune response in the brain, a caveat for this model is that this injection procedure and large amount of viral load within the CNS can cause mechanical damage and viral protein-induced neurotoxicity, which may be independent of viral infection per se. The present study did not assess infection within the brain; however, in the humanized mouse model of HIV-1 infection, we have previously observed CNS immune response following peripheral infection [84]. Here, we identify subtle changes in the NAc due to peripheral EcoHIV infection, which may recapitulate the mild, low-grade neuroimmune activation that characterizes neuroHIV and HAND in PLWH [3,85,86].

The CX3CL1–CX3CR1 signaling axis mediates microglial and neuronal cell communication in pathological conditions including HIV [87,88]. CX3CL1 release by neurons is activity-dependent and inflammatory stimuli drive CX3CL1 release, which consequently regulates microglia activation and exerts neuroprotective effects [89,90]. CX3CL1-CX3CR1 signaling can protect neurons from excitotoxic damage [36,89] and from the toxic effects of HIV protein gp120 [91,92]. CX3CL1 can exert anti-inflammatory effects; it suppresses the release of nitric oxide (NO), IL-6, and tumor necrosis factor (TNF)-α by activated microglia in a dose-dependent manner [37]. CX3CL1 is upregulated in the brain of PLWH with cognitive impairment, without ART therapy [88], consistent with a disease state-dependent effect of CX3CL1-CX3CR1 signaling on CNS outcomes. Contrary to our hypothesis, no significant difference in CX3CL1 levels was observed between sham- and EcoHIV-infected mice. Iba-1 and CX3CL1 were inversely correlated in EcoHIV-infected mice, such that higher levels of Iba-1 were associated with reduced CX3CL1. This relationship was absent in sham mice. It is possible that in EcoHIV, increased CX3CL1-CX3CR1 signaling suppresses microglia activation in the NAc, consistent with previous studies that suggest that microglia activation is negatively regulated by the CX3CL1-CX3CR1 system [89,93].

Importantly, the EcoHIV model does not fully recapitulate HIV-1 infection. One notable difference includes the absence of gp120, which is a known mediator of deleterious immune and neural outcomes in HIV-1 infection [49,94]. While consistent with general peripheral and CNS immune system dysregulation observed in rodent models of HIV [29,45,95], the time course of EcoHIV infection differs from HIV-1 [39]. Here, we report that EcoHIV-infected mice showed viral DNA burdens of up to 0.3–5 × 10^3^ viral DNA copies per 10^6^ spleen cells, with 5 weeks of infection. Treatment with B/F/TAF starting one week following initial infection did not alter EcoHIV viral DNA burden. ART treatment can safely and effectively suppress viraemia to an undetectable level. However, ART does not eliminate HIV DNA [96,97]. EcoHIV appears to establish and maintain a low DNA burden in spleen cells. EcoHIV viral RNA is at its peak viral load within the first week of infection, significantly declining by week 3 post-infection, to undetectable levels at week 8. In contrast, viral DNA levels plateau in the spleen following a transient peak after inoculation, and the DNA burden can remain stable for 15 months after infection [39]. Thus, ART is not necessary for the suppression of EcoHIV viral load within the periphery of wild type mice, indicating an endogenous antiviral immune response that limits EcoHIV replication in mice. Once EcoHIV establishes a chronic infection, the virus persists at a low burden in spleen cells in the form of a latent viral reservoir and ART is not sufficient to restrict EcoHIV viral burden, consistent with our findings. Importantly, it is increasingly apparent that alterations in neurocognitive function do not require persistent elevations in viral load and, further, that the neuroimmune effects of ART are not mediated through suppressing viral load. For example, in a model assessing the contribution of peripheral macrophages migrating to the brain within weeks following systemic EcoHIV infection, ART treatment following infection was unable to attenuate either viral burden or cognitive impairment [39]. Similarly, mild cognitive impairment has been observed in PLWH with controlled HIV load [98]. Together, these findings identify lasting neuroimmune consequences of EcoHIV infection beyond the known windows of high circulating viral loads [30], which are modulated by, but not reversed in the presence of, ART. This represents one potential mechanism by which neurocognitive impairments may develop in the virally suppressed state.

## 5. Conclusions

Together, the findings in these studies identify interactive and independent effects of both EcoHIV infection and ART treatment on the expression of immune targets in the periphery and within the NAc. This underscores the need for preclinical models to emphasize the inclusion of ART as a variable in investigating both systemic and CNS inflammatory outcomes and the potential for ART to alter neuroimmune function.

## Figures and Tables

**Figure 1 cells-13-00882-f001:**
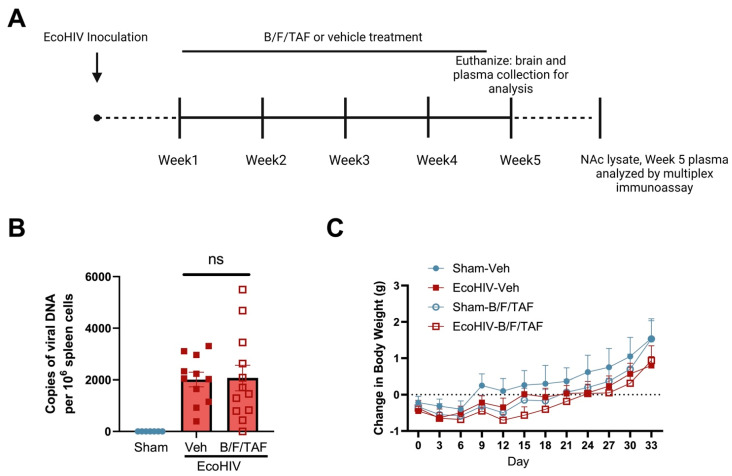
Mouse model of EcoHIV infection. (**A**) Timeline of experiments (created with Biorender.com). Mice were matched based on sex into EcoHIV or sham control groups and for treatment with B/F/TAF. (**B**) Copies of viral DNA at the euthanasia timepoint (week 5 of infection). The number of copies of viral DNA per 10^6^ spleen cells did not differ between EcoHIV-infected mice that were treated with vehicle (n = 11) versus B/F/TAF (n = 12). No DNA copies were detected in the seven representative sham mice. (**C**) Body weight changes versus baseline weight throughout 5 weeks of the experiment. No effects of EcoHIV infection or B/F/TAF treatment were observed on weight change. A main effect of day was observed, such that weight increased over time. Circles and square represent sham and EcoHIV mice; open and closed symbols represent B/F/TAF- and vehicle-treated mice, respectively. Bars represent mean ± SEM. n = 11–12/group.

**Figure 2 cells-13-00882-f002:**
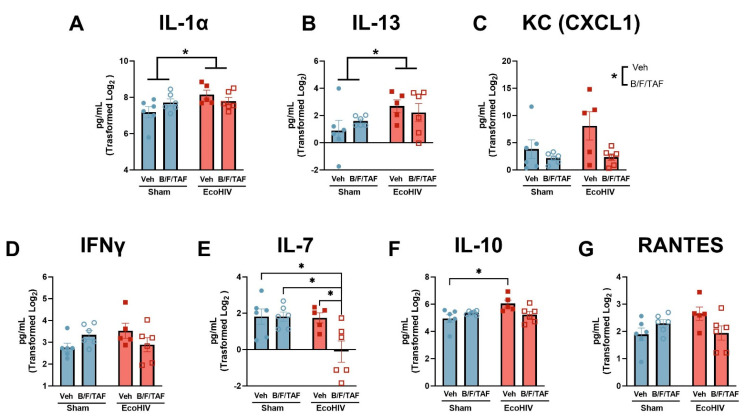
EcoHIV infection and B/F/TAF impacted immune factors in the NAc. EcoHIV, but not B/F/TAF treatment, increased the expression of (**A**) IL-1α and (**B**) IL-13 in the NAc. (**C**) B/F/TAF treatment reduced expression of CXCL1/KC in the Nac, independent of EcoHIV infection. EcoHIV and B/F/TAF interacted to impact (**D**) IFNγ, (**E**) IL-7, (**F**) IL-10, and (**G**) RANTES expression levels, with selective reductions in IL-7 observed in B/F/TAF-treated mice with EcoHIV, and increased IL-10 expression levels only in vehicle-treated EcoHIV-infected mice. Square and circle symbols represent sham and EcoHIV mice; close and open symbols represent vehicle- and B/F/TAF-treated mice, respectively. Bars represent mean ± SEM. n = 5–6/group. * *p* < 0.05.

**Figure 3 cells-13-00882-f003:**
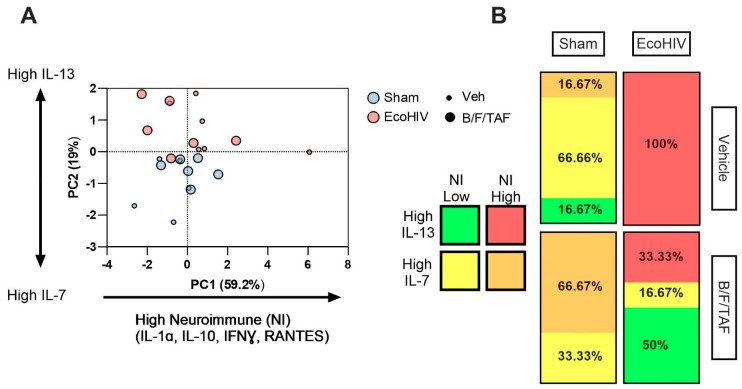
Principal component analysis (PCA) indicated distinct NAc immune profiles based on EcoHIV infection and B/F/TAF treatment. (**A**) Distribution along the first principal component (PC1) was driven by IL-1α, IFNγ, IL-10, and RANTES (“Neuroimmune Cluster”), explaining 52.9% of the cumulative variance. The second principal component (PC2) accounted for 19% of the variance and was positively correlated with IL-13, but negatively correlated with IL-7 expression. Sham vs. EcoHIV status is indicated by color. Vehicle vs. B/F/TAF treatment is indicated by symbol size. (**B**). The expression profiles of EcoHIV-infected and sham control mice are minimally overlapping, with distribution along the PC2 axis corresponding with EcoHIV infection status, with EcoHIV infection associated with High IL-13 and sham associated with High IL-7. Distribution along PC1 corresponded with B/F/TAF treatment and treatment shifted the immune profile to lower neuroimmune factor expression in EcoHIV-infected mice. n = 5–6/group.

**Figure 4 cells-13-00882-f004:**
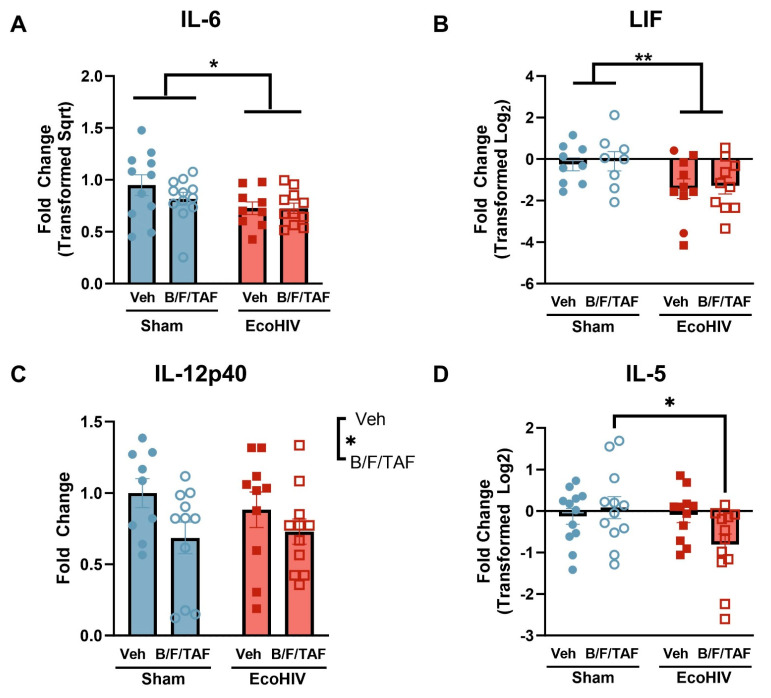
EcoHIV infection and B/F/TAF impacted immune factors in week 5 plasma. EcoHIV infection suppressed the expression of (**A**) IL-6 and (**B**) LIF in the plasma, independent of B/F/TAF treatment. (**C**) B/F/TAF treatment reduced IL-12p40 expression in the plasma of mice, independent of EcoHIV infection. (**D**) EcoHIV and B/F/TAF interacted to alter plasma IL-5 expression, such that IL-5 was reduced only in EcoHIV-infected mice treated with B/F/TAF. Bars represent mean ± SEM. Square and circle symbols represent sham and EcoHIV mice; closed and open symbols represent vehicle- and B/F/TAF-treated mice, respectively. n = 11–12/group. * *p* < 0.05, ** *p* < 0.01.

**Figure 5 cells-13-00882-f005:**
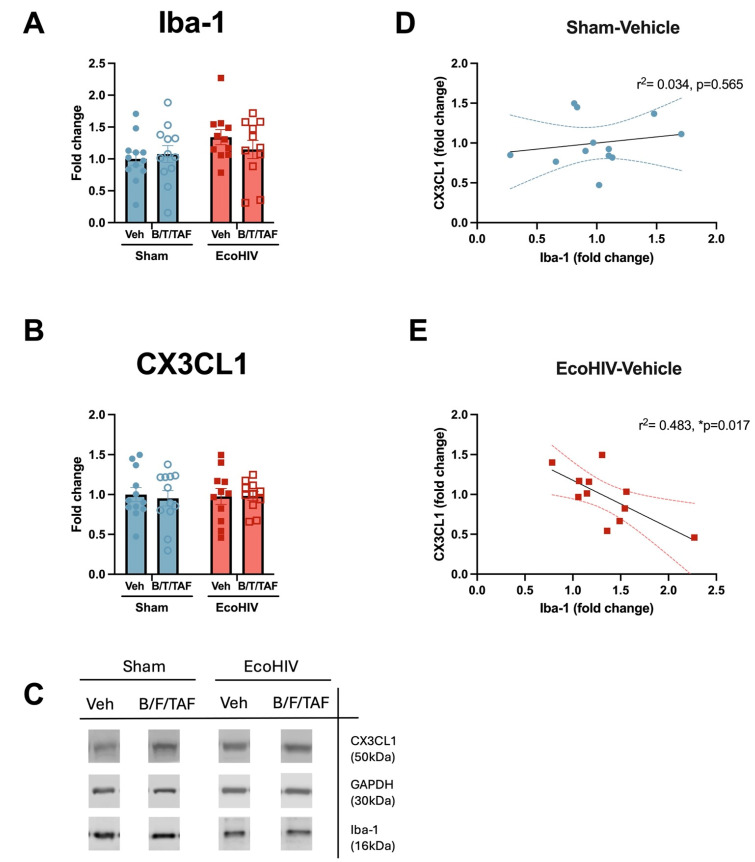
EcoHIV infection determined the relationship between Iba-1 and NAc expression of immune factors. When compared using an ANOVA, no effect of EcoHIV infection or B/F/TAF was observed on NAc (**A**) Iba-1 expression or (**B**) CX3CL1 expression. (**C**) Representative Iba-1, CX3CL1, and internal control GAPDH Western blots. Based on an a priori hypotheses, the overall mean NAc expression of Iba-1 and CX3CL1 were compared between vehicle-treated, Sham- and EcoHIV-infected mice, then simple linear regression was performed between Iba-1 and CX3CL1 in sham-Veh and EcoHIV-Veh mice, respectively. There was no relationship between Iba-1 and CX3CL1 expression in sham mice (**D**), while expression was negatively correlated in EcoHIV-infected mice (**E**). n = 11–12/group. * *p* < 0.05.

**Table 1 cells-13-00882-t001:** Comparison of NAc analyte levels.

Analyte	Effect of EcoHIV		Effect of B/F/TAF		Interaction	
**G-CSF**	**F (1, 19) = 0.01514**	*** p* = 0.9034**	F (1, 19) = 0.9265	*p* = 0.3479	F (1, 19) = 0.1940	*p* = 0.6646
**IFNγ**	F (1, 19) = 0.01158	*p* = 0.9154	F (1, 19) = 0.3436	*p* = 0.5647	**F (1, 19) = 5.144**	*** *p* = 0.0352**
**IL-1α**	**F (1, 19) = 4.472**	*** *p* = 0.0479**	F (1, 19) = 0.1221	*p* = 0.7306	F (1, 19) = 3.315	*p* = 0.0845
IL-1β	F (1, 19) = 4.013	*p* = 0.0596	F (1, 19) = 1.892	*p* = 0.1850	F (1, 19) = 0.0001829	*p* = 0.9894
IL-2	F (1, 19) = 0.04298	*p* = 0.8380	F (1, 19) = 0.0005031	*p* = 0.9823	F (1, 19) = 0.9751	*p* = 0.3358
IL-3	F (1, 19) = 1.982	*p* = 0.1753	F (1, 19) = 2.083	*p* = 0.1652	F (1, 19) = 0.5338	*p* = 0.4739
IL-4	F (1, 16) = 1.172	*p* = 0.2951	F (1, 16) = 2.024	*p* = 0.1741	F (1, 16) = 0.7039	*p* = 0.4138
IL-6	F (1, 19) = 1.474	*p* = 0.2396	F (1, 19) = 0.3504	*p* = 0.5609	F (1, 19) = 2.145	*p* = 0.1594
**IL-7**	**F (1, 19) = 5.683**	*** *p* = 0.0277**	**F (1, 19) = 4.893**	*** *p* = 0.0394**	**F (1, 19) = 4.917**	*** *p* = 0.0390**
IL-9	F (1, 19) = 0.1922	*p* = 0.6660	F (1, 19) = 0.7659	*p* = 0.3924	F (1, 19) = 0.8635	*p* = 0.3644
**IL-10**	**F (1, 19) = 4.414**	*** *p* = 0.0492**	F (1, 19) = 0.7504	*p* = 0.3972	**F (1, 19) = 7.397**	*** *p* = 0.0136**
**IL-13**	**F (1, 19) = 4.520**	*** *p* = 0.0468**	F (1, 19) = 0.04333	*p* = 0.8373	F (1, 19) = 1.096	*p* = 0.3083
IL-12p70	F (1, 19) = 0.002465	*p* = 0.9609	F (1, 19) = 0.4221	*p* = 0.5237	F (1, 19) = 0.1471	*p* = 0.7055
IL-15	F (1, 19) = 0.1649	*p* = 0.6892	F (1, 19) = 0.3582	*p* = 0.5566	F (1, 19) = 1.902	*p* = 0.1839
IP-10	F (1, 19) = 0.04032	*p* = 0.8430	F (1, 19) = 1.502	*p* = 0.2353	F (1, 19) = 0.5769	*p* = 0.4569
**KC**	F (1, 19) = 2.222	*p* = 0.1525	**F (1, 19) = 6.170**	*** *p* = 0.0225**	F (1, 19) = 1.919	*p* = 0.1820
LIF	F (1, 19) = 1.954	*p* = 0.1783	F (1, 19) = 0.0005130	*p* = 0.9822	F (1, 19) = 0.2339	*p* = 0.6342
LIX	F (1, 15) = 0.006022	*p* = 0.9392	F (1, 15) = 0.3435	*p* = 0.5665	F (1, 15) = 0.002027	*p* = 0.9647
MCP-1	F (1, 19) = 2.944	*p* = 0.1024	F (1, 19) = 1.126	*p* = 0.3020	F (1, 19) = 0.02668	*p* = 0.8720
M-CSF	F (1, 19) = 0.1276	*p* = 0.7249	F (1, 19) = 0.4704	*p* = 0.5011	F (1, 19) = 1.819	*p* = 0.1933
MIG	F (1, 19) = 0.1602	*p* = 0.6934	F (1, 19) = 0.4307	*p* = 0.5195	F (1, 19) = 1.887	*p* = 0.1855
MIP-2(CXCL2)	F (1, 19) = 1.419	*p* = 0.2482	F (1, 19) = 2.242	*p* = 0.1508	F (1, 19) = 0.03873	*p* = 0.8461
**RANTES**	F (1, 19) = 0.8639	*p* = 0.3643	F (1, 19) = 0.4738	*p* = 0.4995	**F (1, 19) = 6.197**	*** ** * **p** * ** = 0.0222**

* *p* < 0.05.

**Table 2 cells-13-00882-t002:** Comparison of plasma analyte levels.

Analyte	Effect of EcoHIV		Effect of B/F/TAF		Interaction	
Eotaxin	F (1, 43) = 0.05567	*p* = 0.8146	F (1, 43) = 0.2570	*p* = 0.6148	F (1, 43) = 0.4415	*p* = 0.5099
G-CSF	F (1, 43) = 0.4363	*p* = 0.5124	F (1, 43) = 0.02463	*p* = 0.8760	F (1, 43) = 2.397	*p* = 0.1289
IFNγ	F (1, 41) = 0.2911	*p* = 0.5924	F (1, 41) = 0.2352	*p* = 0.6303	F (1, 41) = 2.408	*p* = 0.1284
IL-1α	F (1, 43) = 1.778	*p* = 0.1894	F (1, 43) = 0.1580	*p* = 0.6930	F (1, 43) = 0.3370	*p* = 0.5646
IL-1β	F (1, 37) = 0.0004847	*p* = 0.9826	F (1, 37) = 2.369	*p* = 0.1323	F (1, 37) = 0.04638	*p* = 0.8307
IL-2	F (1, 31) = 2.877	*p* = 0.0999	F (1, 31) = 0.02509	*p* = 0.8752	F (1, 31) = 0.5275	*p* = 0.4731
IL-3	F (1, 42) = 1.768	*p* = 0.1908	F (1, 42) = 0.02417	*p* = 0.8772	F (1, 42) = 0.4318	*p* = 0.5147
IL-4	F (1, 43) = 0.1110	*p* = 0.7407	F (1, 43) = 2.894	*p* = 0.0961	F (1, 43) = 0.03413	*p* = 0.8543
**IL-5**	F (1, 43) = 3.500	*p* = 0.0682	F (1, 43) = 1.202	*p* = 0.2791	**F (1, 43) = 4.080**	*** *p* = 0.0497**
**IL-6**	**F (1, 39) = 4.804**	*** *p* = 0.0344**	F (1, 39) = 0.8515	*p* = 0.3618	F (1, 39) = 0.8436	*p* = 0.3640
IL-9	F (1, 43) = 0.001259	*p* = 0.9719	F (1, 43) = 3.473	*p* = 0.0692	F (1, 43) = 0.4027	*p* = 0.5290
IL-10	F (1, 43) = 0.9614	*p* = 0.3323	F (1, 43) = 1.168	*p* = 0.2858	F (1, 43) = 0.5482	*p* = 0.4631
**IL-12p40**	F (1, 37) = 0.1134	*p* = 0.7382	**F (1, 37) = 4.728**	*** *p* = 0.0361**	F (1, 37) = 0.5555	*p* = 0.4608
IL-13	F (1, 43) = 2.205	*p* = 0.1449	F (1, 43) = 0.8116	*p* = 0.3727	F (1, 43) = 0.1694	*p* = 0.6827
IL-12p70	F (1, 37) = 0.01623	*p* = 0.8993	F (1, 37) = 0.05424	*p* = 0.8171	F (1, 37) = 0.1065	*p* = 0.7460
IL-15	F (1, 42) = 0.8189	*p* = 0.3707	F (1, 42) = 0.003764	*p* = 0.9514	F (1, 42) = 0.4357	*p* = 0.5128
IL-17	F (1, 43) = 3.818	*p* = 0.0572	F (1, 43) = 0.1825	*p* = 0.6713	F (1, 43) = 0.1935	*p* = 0.6622
IP-10	F (1, 43) = 2.623	*p* = 0.1127	F (1, 43) = 0.03397	*p* = 0.8546	F (1, 43) = 0.08736	*p* = 0.7690
KC	F (1, 43) = 0.1182	*p* = 0.7326	F (1, 43) = 0.09107	*p* = 0.7643	F (1, 43) = 0.4242	*p* = 0.5183
**LIF**	**F (1, 33) = 7.941**	**** *p* = 0.0081**	F (1, 33) = 0.1217	*p* = 0.7294	F (1, 33) = 0.0001909	*p* = 0.9891
LIX	F (1, 43) = 0.1899	*p* = 0.6652	F (1, 43) = 0.06386	*p* = 0.8017	F (1, 43) = 0.07260	*p* = 0.7889
MCP-1	F (1, 40) = 0.05586	*p* = 0.8144	F (1, 40) = 1.063	*p* = 0.3087	F (1, 40) = 0.1222	*p* = 0.7285
M-CSF	F (1, 43) = 1.455	*p* = 0.2344	F (1, 43) = 0.9867	*p* = 0.3261	F (1, 43) = 0.06422	*p* = 0.8012
MIG	F (1, 43) = 3.117	*p* = 0.0846	F (1, 43) = 0.1062	*p* = 0.7461	F (1, 43) = 0.02631	*p* = 0.8719
MIP-1α	F (1, 39) = 0.03899	*p* = 0.8445	F (1, 39) = 3.739	*p* = 0.0605	F (1, 39) = 0.4157	*p* = 0.5229
MIP-1β	F (1, 43) = 0.8469	*p* = 0.3626	F (1, 43) = 0.03914	*p* = 0.8441	F (1, 43) = 0.6362	*p* = 0.4295
MIP-2(CXCL2)	F (1, 32) = 0.1191	*p* = 0.7322	F (1, 32) = 1.226	*p* = 0.2764	F (1, 32) = 0.1690	*p* = 0.6837
RANTES	F (1, 42) = 0.02841	*p* = 0.8670	F (1, 42) = 0.3770	*p* = 0.5425	F (1, 42) = 1.226	*p* = 0.2746
TNFα	F (1, 34) = 0.2426	*p* = 0.6255	F (1, 34) = 0.02013	*p* = 0.8880	F (1, 34) = 0.2063	*p* = 0.6525
VEGF	F (1, 43) = 1.382	*p* = 0.2462	F (1, 43) = 0.004817	*p* = 0.9450	F (1, 43) = 0.2223	*p* = 0.6397

* *p* < 0.05. ** *p* < 0.01.

**Table 3 cells-13-00882-t003:** Correlations between Iba-1 expression and NAc analytes.

	Sham-Veh		EcoHIV-Veh		Sham-B/F/TAF		EcoHIV-B/F/TAF	
	Pearson’s r	*p* Value	Pearson’s r	*p* Value	Pearson’s r	*p* Value	Pearson’s r	*p* Value
M-CSF	−0.068222718	0.8978	0.839266527	# 0.0755	0.117159662	0.8251	−0.206332678	0.6949
MIG	−0.44952674	0.3711	0.817740051	# 0.0908	0.389758049	0.445	0.824767601	*** 0.0434**
LIF	−0.177022696	0.7372	0.745987306	0.1477	−0.512975355	0.298	−0.682293114	0.1354
IL-4	−0.579415488	0.3059	0.728241495	0.163	−0.074996179	0.8877	−0.892826859	0.1072
KC	−0.026069555	0.9609	0.589752177	0.2952	−0.010616142	0.9841	−0.118709927	0.8228
MCP-1	−0.412394917	0.4165	0.375521109	0.5334	−0.894373728	*** 0.0161**	−0.928418899	**** 0.0075**
G-CSF	0.206900311	0.6941	0.35096456	0.5625	−0.426215677	0.3994	−0.403295956	0.4279
IL-13	0.511634468	0.2995	0.225110668	0.7158	−0.086223279	0.871	0.736262156	0.0952
IL-2	0.404048716	0.4269	0.122146027	0.8449	−0.379897186	0.4576	−0.091448348	0.8632
IL-7	−0.390715732	0.4437	0.117940131	0.8502	−0.47748701	0.3382	0.054385453	0.9185
IL-3	−0.00745248	0.9888	−0.166397688	0.7891	−0.117165655	0.8251	−0.38481192	0.4513
IL-6	0.458129801	0.3609	−0.220218491	0.7219	0.474829516	0.3413	0.511465649	0.2997
IL-9	−0.867881407	*** 0.025**	−0.285552826	0.6414	−0.884342292	*** 0.0193**	−0.833558575	*** 0.0392**
RANTES	0.203542734	0.6989	−0.400252342	0.5043	−0.277697902	0.5942	−0.787683252	# 0.0628
IL-15	−0.420596675	0.4063	−0.407445935	0.496	−0.620689799	0.1885	−0.629115677	0.1808
IFNγ	−0.201985846	0.7011	−0.461319212	0.4342	−0.786791051	# 0.0633	−0.80836188	# 0.0516
IP-10	0.210078974	0.6895	−0.59978229	0.285	−0.339432646	0.5104	−0.783924787	# 0.065
IL-10	0.611275269	0.1973	−0.607840565	0.2768	−0.060485338	0.9094	−0.728361676	0.1007
IL-1α	0.398494481	0.4339	−0.647407914	0.2376	−0.587864175	0.2198	−0.955432251	**** 0.0029**
IL-12p70	−0.746521652	# 0.0882	−0.689806805	0.1975	−0.937797428	**** 0.0057**	−0.91026093	*** 0.0117**
MIP-2	−0.472472412	0.344	−0.741169537	0.1518	0.7911121	# 0.0609	0.294744877	0.5707
IL-1β	0.074942205	0.8878	−0.8025595	0.1021	−0.401488171	0.4301	−0.564794703	0.2429
CX3CL1	0.796233045	# 0.0581	−0.959630735	**** 0.0097**	0.712478158	0.1121	0.58665987	0.221

* *p* < 0.05; ** *p* < 0.01; # *p* < 0.1.

## Data Availability

The data supporting the conclusions of this article will be made available upon request.

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
