# Peer review of "Effects of Antiretroviral Treatment on Central and Peripheral Immune Response in Mice with EcoHIV Infection"

_cells, 2024, doi:10.3390/cells13100882_

Round 1

Reviewer 1 Report

Comments and Suggestions for Authors

This is a potentially interesting study that is premature in its current form.  The study appears to have been conducted once; this is inadequate for interpretation and not justified by the experimental system.  Specific comments include:

1.  The study requires demonstration of the effects of the antiretroviral cocktail upon EcoHIV replication as administered to mice-or at a minimum-in culture.  Given that the authors report no effects upon EcoHIV burden in a single compartment-it remains unproven that the drug administration in liquid diets has any relevant effect.

2.  The study relies upon a commercial Elisa for measurement of cytokines in plasma and nucleus accumbens extracts.  Unfortunately, the method for tissue preparation is incorrect judging from the vendor.  They caution against using SDS for tissue solubilization - or at most 0.1%.  The authors employ 1%.

3.  No rationale is presented for antiretroviral treatment beginning one week after virus infection.  EcoHIV continues to enter the brain at this point, obscuring the antiretroviral effects whether confined to prevention of initial infection or upon established infection which is apparently the goal of the study to reproduce effects upon PWH upon antiretroviral therapy.

4.  The study is based almost entirely upon Elisa results from plasma and N Ac extracts.  Only a single Western blot of proteins is shown for confirmation.

5.  The Discussion appears lengthy based upon the modest new findings reported.

Reviewer 2 Report

Comments and Suggestions for Authors

Xie et al. assessed the effects of antiretroviral treatment on central and peripheral immune response in mice with EcoHIV infection. First, the authors confirmed the successful EcoHIV infection with or without antiretroviral treatment in the spleen and then independently examined multiple immune targets in NAc and plasma. The results suggest that EcoHIV infection and ART treatment affect immune responses interactively or independently in the periphery and within the NAc.

The CNS has been indicated as a sanctuary for HIV due to the presence of the blood-brain barrier (BBB), which controls the entry of immune clearance mechanisms and antiretroviral drugs (ARVs). Due to the poor accessibility of human CNS tissues, utilizing the EcoHIV-infected mice as a model to study the immune response in the brain is fascinating, although it couldn’t fully simulate HIV infection. The manuscript is straightforward and well-written. Several questions have been raised, as outlined below:

1.      Did the authors detect viral DNA copies in NAc? I am curious whether EcoHIV could invade the brain in this system. If not, how do you explain the changes in immune markers in NAc?

2.      The authors analyzed immune factors separately in NAc and plasma. Is it possible to compare the expression levels of each immune marker in each group between NAc and plasma? By doing this, we may know which of the brain (NAc) and the periphery (plasma) have the more robust immune response or where the antiretroviral treatment has more effects on the immune response. 

3.      In Materials and Methods, Blood samples were collected from all mice 1, 3, and 5 weeks following EcoHIV inoculation. Clarify the timepoint for plasma samples in Figure 4.

4.      Note the subject number “n” in each Figure.

5.      In Figure 5C, the expression of CX3CL1 is vague. A complete WB with better resolution is required. Please annotate which bands are CX3CL1, Iba-1, and GAPDH in Supplemental Figure 4.

6.      Figure 5D is repetitive with Figure 5A and 5B.

7.      The discussion is comprehensive but unnecessarily too long. Please simplify it.

Round 2

Reviewer 1 Report

Comments and Suggestions for Authors

The authors have responded adequately to several concerns of the original review.  There remain two concerns that can be addressed by revision of the text. 

1.  First, the key goal of the study appears to be demonstration of adverse effects of ART upon host gene expression independent of its antiretroviral effects.  Because no antiretroviral effects were demonstrated, an annoying, but pertinent question arises of the quality of the ART agents.  That is, how does the reader know that the ART is not expired and toxic.  It is sufficient simply to note that the agents are within efficacy span by the manufacturer.

2.  Second, the authors cite the Gu et al (2018) study incorrectly and this affects their view of the mechanism of gene dysregulation by ART.  HIV RNA was detected at 100s to 1000s of copies per microgram RNA up to 3 months after infection, depending upon compartment; p24 elicited Ab responses 8-12 months after infection.  This statement" This is of particular interest as viral RNA levels are expected to be undetectable in EcoHIV-infected mice at this time point, even in the absence of ART (Gu et al., 2018), thus suggesting an independent impact of ART on immune profile in the context of infection."  is incorrect and requires deletion or rephrasing to eliminate the Gu reference.

Author Response

We thank the reviewer for their thoughtful evaluation of our revised manuscript. The revised version addresses the points as described below:

  1. First, the key goal of the study appears to be demonstration of adverse effects of ART upon host gene expression independent of its antiretroviral effects.  Because no antiretroviral effects were demonstrated, an annoying, but pertinent question arises of the quality of the ART agents.  That is, how does the reader know that the ART is not expired and toxic.  It is sufficient simply to note that the agents are within efficacy span by the manufacturer.

We agree that this is an important consideration. We have added the following statement to the text in the Methods:

Compounds were stored and prepared in a manner consistent with manufacturer guidance and were used within the expected efficacy span.

2.  Second, the authors cite the Gu et al (2018) study incorrectly and this affects their view of the mechanism of gene dysregulation by ART.  HIV RNA was detected at 100s to 1000s of copies per microgram RNA up to 3 months after infection, depending upon compartment; p24 elicited Ab responses 8-12 months after infection.  This statement" This is of particular interest as viral RNA levels are expected to be undetectable in EcoHIV-infected mice at this time point, even in the absence of ART (Gu et al., 2018), thus suggesting an independent impact of ART on immune profile in the context of infection."  is incorrect and requires deletion or rephrasing to eliminate the Gu reference.

We regret any issues in our interpretation. We have removed this statement from the manuscript for clarity.